# Cigarette Smoke Induces the Risk of Metabolic Bone Diseases: Transforming Growth Factor Beta Signaling Impairment via Dysfunctional Primary Cilia Affects Migration, Proliferation, and Differentiation of Human Mesenchymal Stem Cells

**DOI:** 10.3390/ijms20122915

**Published:** 2019-06-14

**Authors:** Romina H. Aspera-Werz, Tao Chen, Sabrina Ehnert, Sheng Zhu, Theresa Fröhlich, Andreas K. Nussler

**Affiliations:** Siegfried Weller Research Institute, Department of Trauma and Reconstructive Surgery, Eberhard Karls University Tübingen, BG Trauma Center Tübingen, 72076 Tübingen, Germany; zzuchentao@yahoo.com (T.C.); sabrina.ehnert@gmail.com (S.E.); zhusheng8686@gmail.com (S.Z.); theresa-marie.froehlich@student.uni-tuebingen.de (T.F.); andreas.nuessler@gmail.com (A.K.N.)

**Keywords:** cigarette smoke, TGF-β signaling, MSCs, smokers, fracture, primary cilia, bone metabolic diseases, osteoporosis, Smad signaling

## Abstract

It is well established that smoking has detrimental effects on bone integrity and is a preventable risk factor for metabolic bone disorders. Following orthopedic surgeries, smokers frequently show delayed fracture healing associated with many complications, which results in prolonged hospital stays. One crucial factor responsible for fracture repair is the recruitment and differentiation of mesenchymal stem cells (MSCs) at early stages, a mechanism mediated by transforming growth factor β (TGF-β). Although it is known that smokers frequently have decreased TGF-β levels, little is known about the actual signaling occurring in these patients. We investigated the effect of cigarette smoke on TGF-β signaling in MSCs to evaluate which step in the pathway is affected by cigarette smoke extract (CSE). Single-cell-derived human mesenchymal stem cell line (SCP-1 cells) were treated with CSE concentrations associated with smoking up to 20 cigarettes a day. TGF-β signaling was analyzed using an adenovirus-based reporter assay system. Primary cilia structure and downstream TGF-β signaling modulators (Smad2, Smad3, and Smad4) were analyzed by Western blot and immunofluorescence staining. CSE exposure significantly reduced TGF-β signaling. Intriguingly, we observed that protein levels of phospho-Smad2/3 (active forms) as well as nuclear translocation of the phospho-Smad3/4 complex decreased after CSE exposure, phenomena that affected signal propagation. CSE exposure reduced the activation of TGF-β modulators under constitutive activation of TGF-β receptor type I (ALK5), evidencing that CSE affects signaling downstream of the ALK5 receptor but not the binding of the cytokine to the receptor itself. CSE-mediated TGF-β signaling impaired MSC migration, proliferation, and differentiation and ultimately affected endochondral ossification. Thus, we conclude that CSE-mediated disruption of TGF-β signaling in MSCs is partially responsible for delayed fracture healing in smokers.

## 1. Introduction

Cigarette smoking (CS) continues to be the leading global cause of preventable death. In 2030, CS will cause 8 million deaths per year worldwide, according to the World Health Organization. Tabaco consumption denotes a major health risk that affects the entire human body and is linked to various health disorders, including coronary heart disease, chronic obstructive pulmonary disease, cerebrovascular disease, and cancer [1,2]. 

Interestingly, CS also affects bone integrity, with a positive correlation between years of exposure and the number of cigarettes smoked per day [1,3]. Furthermore, smokers submitted to orthopedic surgery have an increased risk of delayed fracture healing, complications (e.g., infections and non-union fractures), and longer hospital stays [1,3]. CS is considered one of the main social risk factors for developing metabolic bone diseases [3,4,5,6]. Metabolic bone diseases describe a diverse group of pathologies (e.g., osteoporosis, Paget disease, Rickets, osteomalacia, diabetic osteopathy) that impair bone remodeling for different reasons: impaired bone formation by osteoblasts, enhanced bone degradation by osteoclasts, or a combination of both. Osteoporosis is the most prevalent metabolic bone disease, characterized by decreased bone strength and increased risk of fractures [7]. CS significantly increases the likelihood of developing osteoporosis [8].

Fractures in patients suffering from metabolic bone diseases are associated with delayed or non-union fracture healing. In these patients there is an imbalance in bone-forming and bone-resorbing cells [9,10]. Transforming growth factor β (TGF-β) is a multifactorial regulatory protein that has several effects on mesenchymal stem cells (MSC), e.g., migration, proliferation, and differentiation [11,12]. 

Once TGF-β is released from the bone matrix, the signaling pathway commences with binding of the ligand to the TGF-β receptor complex. This action activates the canonical (Smad-dependent) TGF-β signaling pathway, which regulates the transcription of several target genes [13]. The TGF-β receptor complex can also activate a non-canonical, Smad-independent pathway that involves other factors, such as mitogen-activated protein kinase pathways (extracellular-signal-regulated kinases, c-Jun N-terminal kinase, and p38), Rho-like Guanosine triphosphate hydrolase enzymes signaling pathways, or phosphatidylinositol-3-kinase/Protein kinase B pathways [14]. 

Besides the direct stimulation of MSCs, osteoblasts, and chondrocytes, and inhibition of osteoclasts, TGF-β can boost the effect of other factors, like bone morphogenetic proteins (BMPs) and insulin-like growth factors that assist in fracture healing [15,16,17].

Canonical TGF-β signaling is reportedly partially controlled by the microtubule base organelle (primary cilia) in stem cells that differentiate into cardiomyocytes; proper downstream activation is reliant on clathrin-dependent endocytosis at the cilia pocket region [18]. This organelle can coordinate the activity of multiple signaling pathways during tissue development and homeostasis [19]. Mutations, as well as small interfering RNA (siRNA) that affect the intraflagellar transport system (IFT) in primary cilia, impair signaling and the primary cilia structure [20,21]. Mutations in the *IFT88* gene reduce TGF-β-mediated Smad2/3 activation, results that demonstrate the primary cilia structure is indispensable for the correct functioning of the pathway [22]. Additionally, depleted *IFT88* in MSCs reduces TGF-β-induced migration [23]. Our previous studies demonstrated that exposure to cigarette smoke extract (CSE) affects osteoblast function and impairs MSC osteogenic differentiation. Interestingly, CSE exposure also affects the primary cilia structure in these cells during differentiation [24,25,26,27,28].

Surprisingly, smokers present lower serum TGF-β concentrations than non-smokers [13,29]. After a fracture, TGF-β levels increase during endochondral ossification in order to attract MSCs to form the cartilage callus, which is later systematically replaced with mineralized tissue by differentiated MSCs [13]. At this stage, smokers show a positive correlation between decreased TGF-β levels and delayed fracture healing [13,15]. 

However, it is still not known how CS affects the TGF-β signaling pathway. Therefore, the purpose of this study was to elucidate the effects of CSE on TGF-β signaling and how it influences the migration, proliferation, and appropriate differentiation of MSCs.

## 2. Results

### 2.1. CSE Downregulated TGF-β Signaling Through Disruption of Primary Cilia on SCP-1 Cells

Previous studies revealed that exposure to CSE disrupts the primary cilia structure and therefore impairs the osteogenic differentiation of the human telomerase reverse transcriptase immortalized single-cell human mesenchymal cell line (SCP-1 cells) [24,25]. 

SCP-1 cells infected with an adenoviral-based reporter construct (Ad5-CAGA9-MLP-Luciferase) were exposed to CSE for 24 h, followed by induction of the TGF-β pathway with rhTGF-β1. These cells exhibited a dose-dependent reduction in TGF-β signaling; there was statistical significance at 10% *v/v* CSE (Figure 1a). Induction of Smad2/3 signaling was evaluated by measuring luciferase activity in protein lysates from SCP-1 cells.

To emphasize the role of primary cilia on TGF-β signaling, we also investigated the effect of the chemical disruption of primary cilia on TGF-β signaling. SCP-1 cells treated with chloral hydrate (CH, 0.5–1 µM) showed disrupted primary cilia structure (Figure 1c–e), a result that confirmed earlier published results with CSE [24,25]. Following the same line of evidence, pharmacological disruption of primary cilia significantly reduced TGF-β signaling (Figure 1b). However, TGF-β signaling was not entirely abolished after primary cilia disruption, a finding that evidenced receptors located in this organelle contributed to the pathway, but receptors localized in the membrane also activated the basal TGF-β pathway.

### 2.2. Protection of Primary Cilia Structure with Resveratrol rescues TGF-β Signaling Suppressed by CSE

In order to confirm that the disruption of the primary cilia structure leads to aberrant TGF-β signaling, primary cilia structures were protected from the deleterious effects of CSE with resveratrol. Resveratrol is a polyphenol found in grapes with antioxidant properties [30]. Resveratrol administration in mice exposed to CS reduced cilia loss in trachea epithelia [31]. Moreover, co-incubation with resveratrol protected primary cilia against the deleterious effects of CSE via a reduction of oxidative stress in SCP-1 cells [24].

SCP-1 cells infected with an adenoviral Smad2/3 reporter construct (Ad5-CAGA9-MLP-Luciferase) were co-incubated with resveratrol (1 µM) and CSE for 24 h, followed by the induction of the TGF-β pathway with rhTGF-β1. These cells co-incubated with resveratrol and CSE exhibited an increase in TGF-β signaling in comparison to CSE exposure alone (Figure 2a). To confirm the protective effects of resveratrol on the primary cilia structure, immunofluorescence analysis showed that co-incubation with resveratrol significantly increased the cilia length and the number of ciliated SCP-1 cells (Figure 2b–d). These results support the evidence that dysfunctional primary cilia affect the propagation of TGF-β signaling.

### 2.3. CSE Reduced the Levels of Downstream TGF-β Pathway Mediators and the Nuclear Translocation of the Active Complex

We next investigated the effect of CSE on the protein levels of active TGF-β signaling modulators. SCP-1 cells were exposed to CSE (5% *v/v*) for 14 days in order generate chronic damage in the primary cilia structure. To avoid indirect effects due to cytotoxicity or additive effects of long term exposure to CSE, SCP-1 cells were only treated with 5% *v/v* CSE. After 14 days, TGF-β signaling was induced by the addition of rhTGF-β1 (10 ng/mL) for 1 h. Downstream TGF-β signaling mediator protein expression levels were analyzed by Western blot. rhTGF-β1 increased phospho-Smad2 and phospho-Smad3 levels. However, the induction of phospho-Smad2 was higher than phospho-Smad3, a result that indicates Smad2 more dominantly mediated TGF-β signaling in SCP-1 cells (Figure 3a,b). As expected, CSE decreased the levels of active mediators; phospho-Smad2 downregulation was more pronounced (Figure 3a). Furthermore, CSE significantly reduced Smad4 protein (Figure 2c), a cofactor necessary to assemble the active complex with phospho-Smad2/3, that propagates the signaling complex to the nucleus. Thus, it is feasible to speculate that the concentration of the active complex in the nucleus was reduced. Therefore, we investigated whether the nuclear translocation of the active complex was affected by CSE. SCP-1 cells were treated with CSE (5–10% *v/v*) or CH (0.5–1 mM), and then signaling was induced with rhTGF-β1 (10 ng/mL) for 1 h. Nuclear translocation of the active complex was analyzed by Smad3 immunofluorescence. CSE (5–10% *v/v*) exposure significantly reduced Smad3 nuclear localization after TGF-β signaling induction (Figure 3e,f). This finding suggests that there are not any compensatory mechanisms in the nuclear translocation system that compensate for the lower phospho-Smad2/3 and Smad4 protein levels and consequently propagate the signal to the nucleus. Nuclear localization of the phospho-Smad3/4 complex was also reduced in SCP-1 cells with CH-stunted primary cilia (Figure 3e,f), a result that indicates TGF-β signaling is associated with primary cilia structure. These results suggest that defective primary cilia could lead to aberrant cell signaling coordination under smoking conditions. Possible regulations may be due to failures in receptor–ligand interactions, impaired internalization of the ligand–receptor complex, or affected the kinase receptor function.

### 2.4. CSE Perturbed Normal TGF-β Receptor Type I Function

Since CSE reduced active Smad2/3 protein levels, and consequently reduced nuclear translocation of the active pSmad3/4 complex, we evaluated the effect of CSE on the function of TGF-β receptor kinase type I (ALK5). ALK5 is responsible for phosphorylating and activating TGF-β signaling mediators (e.g., Smad2/3) with its serine/threonine kinase activity. SCP-1 cells were infected with Ad5-caALK5 virus particles. The expressed ALK5 was genetically modified to constitutively activate Smad2/3 phosphorylation and the associated signaling, independent of TGF-β binding to the receptor. Protein expression levels of phospho-Smad2 were evaluated in protein lysates from SCP-1 cells infected with Ad5-caALK5 and treated with CSE (5–10% *v/v*) for 48 h. Constitutive ALK5 activation increased the phospho-Smad2 protein level (Figure 4). The addition of CSE significantly decreased the phospho-Smad2 protein in a dose-dependent manner (Figure 4). Our data clearly suggest that the observed effects could be mediated by failure in the TGF-β mediator activation by the ALK5 receptor and not from unsuccessful binding of the ligand to the receptor.

### 2.5. CSE Affected SCP-1 Cell Migration and Proliferation

MSC migration to the fracture site is an important step for successful healing in patients with delayed fracture healing or non-union fractures due to bone metabolic diseases. TGF-β was proposed as a key chemokine for MSCs, and smokers present lower serum TGF-β levels after fracture than non-smokers [29]. Therefore, we examined the effect of CSE on MSC migration using a scratch assay. A wound was generated in SCP-1 cell monolayers with following exposure to CSE (5–10% *v/v*) and rhTGF-β1 (10 ng/mL). After 16 h, the wound closure was evaluated. SCP-1 cells exposed to CSE (5–10% *v/v*) for 16 h exhibited significantly reduced wound closure (Figure 5a,b), which evidenced that CSE negatively and dose-dependently affected cell migration. However, supplementing the cells with rhTGF-β1 (10 ng/mL) accelerated SCP-1 migration to the wound (Figure 5a,b), a finding that supports the TGF-β chemokine function for MSCs [32]. TGF-β also induces cell proliferation. To directly determine whether CSE affects cell proliferation, SCP-1 cells were treated with CSE (5% *v/v*) for 24 or 48 h, with or without co-incubation with rhTGF-β (10 ng/mL), to induce proliferation. An assessment of the protein expression levels of the proliferation marker proliferating cell nuclear antigen (PCNA) by Western blot indicated that CSE did not affect the proliferation rate after 24 h (Figure 5c,d). However, 48 h CSE exposure decreased the proliferation rate. TGF-β pathway induction increased the PCNA level after 24 h under control conditions. However, this positive effect was not reproduced in cells exposed to CSE (Figure 5c,d). These results indicate that TGF-β directly promoted cell migration under CSE exposure, but TGF-β addition could not compensate for the detrimental SCP-1 cell proliferation caused by CSE.

### 2.6. Impaired TGF-β Signaling by CSE Negatively Affected Osteochondral Progenitor Cell Differentiation

During endochondral bone fracture repair, new bone is formed through a cartilage intermediate (callus) produced from chondrogenicaly differentiated MSCs [33]. Several studies demonstrated that TGF-β plays an essential role during MSC chondrogenic differentiation [34,35,36]. Thus, we investigated whether CSE-modulated TGF-β signaling disruption affected the expression of chondrogenic markers on MSCs differentiated into chondrocytes. SCP-1 cells were exposed to CSE during chondrogenic differentiation. Since after 14 days of chondrogenic differentiation, SCP-1 cells showed a chondrogenic phenotype (positive staining for glycosaminoglycan and proteoglycan (data no shown)), this time the point was selected to determine effects of CSE on this TGF-β mediated mechanism. At day 14, total RNA Semi-quantitative gene expression analysis revealed that *Collagen II*, a major extracellular matrix protein in cartilage, was significantly downregulated in SCP-1 cells exposed to CSE and treated with TGF-β for 14 days (Figure 6a,e). Surprisingly, the expression of *Aggrecan,* a specific marker for the cartilage extracellular matrix, was upregulated in cells exposed to CSE and TGF-β, a result that suggests a compensatory mechanism from the cells in response to decreased Collagen II with CSE treatment (Figure 6d,e). Intriguingly, the hypertrophic chondrogenic phenotype marker *Collagen X* was upregulated with CSE treatment and TGF-β induction (Figure 6b,e). The transcriptional factor *Sox9* induces the differentiation of MSCs into pre-chondrocytes. As expected, TGF-β treatment downregulated this transcription factor. However, the expression of *Sox9* was upregulated in cells exposed to CSE (Figure 6c,e). Following the same line of results, impaired TGF-β signaling via CH-mediated primary cilia disruption exhibited a similar expression pattern to CSE exposure. Taken together, CSE exposure affects the expression of chondrogenic markers, which may disturb MSC chondrogenic differentiation and result in a hypertrophic phenotype. Consequently, these results could explain the altered endochondral ossification observed in smokers during long bone fracture healing [37,38].

## 3. Discussion

Smoking is considered one of the main risk factors for developing metabolic bone diseases [3,4,5,6]. Metabolic bone diseases are characterized by impaired bone remodeling for different reasons: disrupted bone formation by osteoblasts, enhanced bone degradation by osteoclasts, or a combination of both. Fractures in patients suffering from metabolic bone diseases are clearly associated with delayed or non-union fracture healing regarding imbalances between bone-forming and bone-resorbing cells [9,10].

TGF-β1 is a multifunctional signaling protein that significantly affects bone cells and plays an essential role in the maintenance of appropriate bone remodeling [12,13,39,40]. Furthermore, TGF-β induces the expression of extracellular matrix compounds like collagen type I, fibronectin, osteopontin, osteonectin, thrombospondin, and proteoglycans, all of which contribute to fracture healing [13,29,41]. Animal studies have revealed that a lack of TGF-β causes defects in bone strength and microarchitecture. Additionally, the local presence of cytokines (TGF-β, BMP, Insulin-like growth factor 1, etc.) enhances the fracture healing process [42].

Interestingly, smokers present lower serum TGF-β concentrations compared to non-smokers [15,29]. After a fracture, TGF-β increases during the inflammatory processes in order to guide immature progenitor cells to invade the fracture area. At this stage, patients with delayed fracture healing (as well as smokers) exhibit a marked decrease in systemic TGF-β1 levels [15]. However, the effects of CSE on the TGF-β signaling pathway of bone cells are still unclear. The current study demonstrated that CSE directly inhibited canonical TGF-β signaling in MSCs. CSE-mediated downregulation of TGF-β signaling was dose dependent, and the disruption of primary cilia by pharmacological treatments presented similar results. Furthermore, we performed a rescue experiment confirming the central role of the primary cilia structure on TGF-β signaling. We observed that protection of the primary cilia structure from CSE deleterious effects by co-incubation with resveratrol significantly reversed the negative effects on TGF-β signaling. Several studies associated primary cilia with different signaling pathways and postulated that the organelle is a key player in signaling translation [18,22,43]. Although CSE detrimentally affected TGF-β signaling, the pathway was not totally abolished. This fact highlights that receptors localized in the primary cilia propagate signaling activation, but receptors in the cell membrane could also propagate, contributing to the pathway activation.

Since we observed significant CSE-mediated inhibition of canonical TGF-β signaling, we evaluated changes in the levels of active canonical TGF-β signaling downstream modulators (Smad2 and Smad3). CSE exposure reduced phospho-Smad2 and phospho-Smad3 levels. We detected that TGF-β signaling induction had a greater effect on phospho-Smad2, a result that suggests this mediator is more affected by CSE. Smad2 knockout is embryonically lethal in mice, while Smad3-deficient mice are viable. Thus, Smad2 function can apparently compensate for Smad3, but Smad3 cannot compensate for Smad2 [44].

Smad4 is the central cofactor mediator of TGF-β signaling. Smad4 binds to phosphorylated Smads to form the active complex that translocates into the nucleus and triggers target gene transcription. Smad4 is required to maintain normal bone homeostasis [45]. In vitro studies proposed that Smad4 interacts with transcriptional factors, such as Runx2 and AP-1 (c-Fos-JunD), that influence MSC osteogenic differentiation [46]. Moreover, conditional Smad4 knockout in chondrocytes amplifies the hypertrophic phenotype [47]. Fascinatingly, CSE exposure downregulated basal MSC Smad4 expression, which caused the cells to have less available cofactor to form the activated complex, and consequently, the activation was not adequately propagated to the nucleus.

Nuclear translocation of the active complex differs for Smad2 and Smad3. For Smad3/Smad4, the activated complex translocates in an importin-dependent manner (mediated by importin-β1) [48,49] or by associating with nuclear pore proteins (NUP214/NUP153) [50]. However, for Smad2/Smad4, nuclear translocation occurs only through nuclear pore proteins [51]. Since Smad3 has multiple translocation mechanisms, it is an attractive target to investigate in order to elucidate whether there is a compensatory mechanism that balances the lack of phosphorylated Smads under CSE exposure. As expected, CSE-mediated TGF-β signaling disruption impaired Smad3/Smad4 translocation into the nucleus. This result demonstrates that the reduced levels of active protein are not compensated by the nuclear import machinery. 

Next, we investigated whether CSE exposure affected the TGF-β ligand–receptor interaction, since it is a critical step in pathway initiation. A constitutively activated TGF-β receptor type I (ALK5) did not abolish the dose-dependent reduction of phospho-Smad2 induced by CSE. This result suggests that the observed CSE effects were mediated by impaired TGF-β downstream mediator (Smad2/Smad3) activation through ALK5 and not by inhibition of the ligand–receptor interaction. Accurate internalization of the active TGF-β receptor–ligand complex enhances the activation of downstream Smad mediators. Clathrin-dependent endocytosis of the ligand–receptor complex is reduced at the cilia pocket region in cells with truncated primary cilia structure [52]. We suggest that CSE exposure could affect endocytosis of the activated receptor–ligand complex in the cilia pocket region via a defective primary cilia structure, a phenomenon that would lead to aberrant cell signaling coordination by this organelle. Inappropriate endocytosis affects the interaction between TGF-β receptor type I and a cofactor, such as the Smad anchor for receptor activation protein (SARA), which enhances the association with Smad mediators and improves their phosphorylation [53]. It is possible that CSE exposure either affects binding of cofactors (to interfere in proper Smad phosphorylation) or inhibits the serine/threonine kinase function of the receptor type 1.

For bone tissue, TGF-β1 plays a crucial role in MSC migration, regulating osteoblast function (induce proliferation and inhibit late differentiation), appropriate chondrogenic differentiation, and consequently influences the bone healing process [13,32,34,40,41,54,55,56,57,58]. Cell migration is also coordinated by primary cilia via appropriate TGF-β signaling. In this regard, defects in the formation or sensory function of primary cilia are associated with a series of migration-related disorders and diseases [59]. Interestingly, CSE dose dependently reduces MSC migration, as previously reported [60]. However, TGF-β signaling induction through the addition of rhTGF-β1 reversed the detrimental effects of CSE on cell migration. These results suggest that topic TGF-β1 application could enhance MSC recruitment in the fracture place. Nevertheless, TGF-β1 supplementation did not reverse the decreased proliferation (denoted by PCNA expression) observed with 48 h CSE exposure, a finding that challenges the use of rhTGF-β in smokers to enhance fracture healing.

After long bone fracture, MSCs migrate to the fracture, condensate, differentiate into chondrocytes, and begin to produce cartilage. Cartilage is later systematically replaced with mineralized tissue by osteoblasts derived from the recruited MSCs during the process of endochondral ossification [61,62]. Since endochondral ossification is an essential process during long bone fracture healing, and TGF-β regulates the behavior and function of the cells involved in the process [61], we investigated the effect of CSE on MSC chondrogenesis via TGF-β signaling induction. Fourteen-day exposure to TGF-β significantly increased the gene expression of *Collagen II*, a major extracellular matrix protein in cartilage, and downregulated *Sox9*, a marker of pre-chondrocytes, and *Collagen X*, a marker of hypertrophic chondrocytes. Interestingly, CSE dose dependently decreased *Collagen II* expression independent of TGF-β signaling induction. This result demonstrates that the cartilage structure produced by chondrocytes exposed to CSE is feeble. Moreover, the hypertrophic marker *Collagen X* increased with CSE. Increased *Sox9* gene expression (a transcription factor necessary to activate the earliest chondrogenic-specific genes) demonstrated an improper MSC chondrogenic differentiation under CSE exposure. These results are supported by the fact that human adipose-derived MSCs treated with CSE show decreased *Collagen II* and increased *Sox9* [60]. Furthermore, alcian blue staining (which stains glycosaminoglycans in cartilage) is decreased in human periodontal ligament-derived stem cells isolated from smokers [63]. We previously showed that CSE exposure downregulates Smad4 protein levels. Smad4 deletion in chondrocytes causes a disorganized growth plate, reduces chondrocyte proliferation, increases apoptosis, and accelerates hypertrophic differentiation [23]. Similar effects were reported in Smad4 mutant mice, results that demonstrate Smad4 is necessary for maintaining sequential chondrocyte differentiation [47]. The lack of Smad4 could partly explain the impaired CSE-exposed MSC chondrogenic differentiation. Surprisingly, CSE induced Aggrecan expression, which is a major component of the cartilage extracellular matrix. Nevertheless, one study questioned the use of Aggrecan as a chondrocyte marker since it is constitutively expressed by MSCs [64]. Additionally, increased Aggrecan expression may also be associated with chondroid accumulation in response to damage [65]. Moreover, a disruption of the primary cilia from chondroprogenitor cells leads to reduced Collagen II, Collagen X, and BMP-2 expression and cannot be reestablished by mechanical activation. However, the role of primary cilia in hypertrophic chondrocytes is more restricted, since primary cilia disruption only reduces Collagen X in response to mechanical activation. These results demonstrated the role of primary cilia in regulating the chondrogenic profile of progenitor cells [66]. Furthermore, CH-mediated primary cilia disruption impaired TGF-β signaling and downstream activation. This expression pattern was similar to CSE-exposed MSCs, which affect chondrogenic differentiation and led to the hypertrophic differentiation profile. These results partially support the in vivo observations that CSE affected endochondral ossification and consequently delayed or impaired long bone fracture healing. 

Based on our findings, we consider that topic rhTGF-β1 application could provide positive effects at early stages of endochondral ossification among smokers (inducing progenitor cell migration). However, proliferation and appropriate osteochondral progenitor cell differentiation will be compromised due to the block in the canonical TGF-β signaling pathway. Therefore, therapies that enhance or reestablish the canonical pathway or activate non-canonical Smad phosphorylation (e.g., riluzole, which activates glycogen synthase kinase 3 for Smad2/3 activation [67]) may be a more promising alternative for smokers who suffer from delayed fracture healing compared to rhTGF-β1 application.

Additionally, the findings of this study have to be seen in the light of some limitations. The effect of CSE on the canonical TGF-β signaling through primary cilia structure were only evaluated in MSCs. However, during bone fracture, several cells play an important role in the success of the fracture healing. Therefore, CSE’s detrimental effects in TGF-β signaling could also influence the function of immune cells (at early stages of the healing process) or osteoclast (at the remodeling stage of the healing process) [62], whereby, co-culture systems might be useful for screening purposes to study different effects of CSE on various bone cells during fracture healing.

## 4. Materials and Methods 

### 4.1. Materials

Cell Culture Medium and supplements were purchased from Life Technologies (Darmstadt, Germany). Chemicals were obtained from Sigma (Munich, Germany). Recombinant human active TGF-β1 was obtained from Peprotech (London, UK).

### 4.2. SCP-1 Cells Culture and Chondrogenic Differentiation

Human immortalized bone marrow mesenchymal stem cells (SCP-1 cells, provided by Dr. Matthias Schieker [68]) were cultured in Minimum Essential Medium Eagle alpha (MEM α) supplemented with 10% *v/v* fetal bovine serum (FBS), 100 U/mL penicillin and 100 mg/mL streptomycin, in a water-saturated atmosphere of 5% CO_2_ at 37 °C [69]. Chondrogenic differentiation was induced with Dulbecco’s Modified Eagle Medium (DMEM) high glucose medium containing 100 nM Dexamethasone, 1 mM sodium pyruvate, 220 µM L-ascorbic acid-2 phosphate, 347 µM L-proline, 1.25 g/mL bovine serum albumin (BSA), 625 µL liquid media supplement ITS 100x stock (mixture of recombinant human insulin, human transferrin, and sodium selenite), 20 ng/mL linolic acid, 100 U/mL penicillin, and 100 mg/mL streptomycin. Chondrogenic differentiation was confirmed with glycosaminoglycan and proteoglycan positive staining (Alcian blue and Safranin O). Chondrogenic differentiation was adapted from a previous publication with SCP-1 cells [68]. For experiments, 5% CSE, 10% *v/v* CSE, 0.5 mM chloral hydrate, or 10 ng/µL rhTGF-β1 were added to the media. The medium was changed twice a week during chondrogenic differentiation, which was sustained for 14 days.

### 4.3. Cigarette Smoke Extract Generation

The smoke of the combustion of two commercial cigarettes (Marlboro, Philip Morris, New York City, USA), containing 0.8 mg nicotine and 10 mg tar each, was continuously bubbled through a 50 mL pre-warmed MEM α medium (0% *v/v* FCS) in a standard gas wash bottle, at a speed of 95 bubbles/min, as described before [24]. The concentration of CSE was normalized by its optical density at 320 nm (OD320), with an OD320 of 0.7 considered 100% *v/v* CSE [26]. CSE was freshly prepared for every experiment and sterile filtered (0.22 µm pores filter) before diluted to reach 5% and 10% *v/v* CSE, which corresponds to exposures associated with smoking 10 cigarettes (0.5 pack) a day to 20 cigarettes (1pack) a day [70].

### 4.4. Transient SCP-1 Cells Infections and Reporter Assay

For the TGF-β reporter assay, SCP-1 cells were stably infected with an adenoviral vector system expressing luciferase under the control of Smad2/3-responsive element (Ad5-CAGA9-MLP-Luciferase, kindly provided by Professor Peter ten Dijke [71,72]) 1:7 (*v/v*), as described before [73]. After 24 h, the cells were washed with phosphate-buffered saline (PBS) and treated with 0.5–1 mM chloral hydrate (16 h) or 5–10% CSE (24 h). Subsequently, the cells were exposure to 10 ng/mL rhTGF-β1 for 24 h. Upon binding of phosphorylated Smad3/4 (induced by rhTGF-β1), luciferase was expressed by the cells. Cell lysate preparation and luciferase measurement were performed according to the manufacturer’s instructions, using the Steady-Glo Luciferase Assay System (Promega, Madison, USA), and normalized to total protein content, measured with Lowry. In order to investigate TGF-β effects independent of substrate binding, SCP-1 cells were infected with adenoviral particles, resulting in the expression of constitutive active ALK5 (Ad5-caALK5, kindly provided by Professor Peter ten Dijke [74]) 1:50 *v/v*. The expressed ALK5 was genetically modified in a way to constitutively activate Smad2/3 phosphorylation and associated signaling independent of subtract binding. After 5 h, cells were washed with PBS and treated with 5–10% CSE (24 h). Protein expression levels of active Smad2/3 was evaluated by Western blot. Adenovirus infection efficiency was >90%, as shown by the fluorescent microscopy of cells infected with Ad5-green fluorescent protein (GFP) after 24 h (Appendix A
Figure A1) [73]. 

### 4.5. Inmunofluorescence Staining

After treatment, cells were washed with Dulbecco’s phosphate-buffered saline (DPBS) and fixed with 4% (*w/v*) paraformaldehyde for 10 min at room temperature. This was followed by permeabilization with 0.2% Triton-X-100 solution for 20 min at room temperature and treatment with 2% (*w/v*) paraformaldehyde for 10 min at room temperature. Unspecific binding sites were blocked with 5% (*w/v*) BSA for 1 h at room temperature, followed by incubation with the first antibody overnight at 4 °C (Table 1). After washing three times with PBS, cells were incubated with Alexa-fluor labeled secondary antibody (1:1000) for 2 h at room temperature (Table 1). Nuclei were stained with Hoechst 33,342 (1:1000). Images were taken with an epifluorescence microscope (EVOS FL, life technologies, Darmstadt, Germany). Pictures were analyzed with the ImageJ software (Version 1.5, NIH, Bethesda, MD, USA) by two independent investigators in a blinded fashion. Based on the microscopic pictures taken, cilia length was determined by the maximum intensity projection method [58].

### 4.6. Western Blot Analysis

Cells were lysed in a freshly prepared ice-cold radioimmunoprecipitation assay buffer (RIPA) with protease and phosphatase inhibitors. After quantification with micro Lowry, 30 μg total protein was separated by sodium dodecyl sulfate–polyacrylamide gel (SDS page) and transferred to nitrocellulose membranes. Membranes were blocked with 5% *v/v* BSA in Tris-buffered saline with Tween20 (TBS-T) for 1 h at room temperature. After overnight incubation with primary antibodies in TBS-T (1:1.000) at 4 °C, membranes were incubated with the corresponding peroxidase-labeled secondary antibodies in TBS-T (1:10.000) for 2 h at room temperature (Table 1). For signal development, membranes were incubated for 1 min with an electrogenerated chemiluminescence (ECL) substrate solution. Chemiluminescent signals were detected by a charge-couple device camera (INTAS Science Imaging, Göttingen; Germany) and quantified using the ImageJ software [25].

### 4.7. SCP-1 Cells Migration Assay—Scratch Assay

SCP-1 cells were plated at high density in 24-well plates. After 24 h, the cell monolayer was mechanically wounded with a 200 µL pipet tip. Immediately after setting the wound, the medium was changed to remove detached cells and start stimulation with 5–10% CSE. The “scratches” were documented by taking microscopic images directly after wounding (0 h) and after 16 h. For better visualization, cells were stained with Sulforhodamine B (SRB) [25]. Wound closure was quantified with the ImageJ software by using the following formula: 100 − (area16h × 100/area0h) [32].

### 4.8. Semi-Quantitative Reverse-Transcription Polymerase Chain Reaction RT-PCR

Total RNA was isolated from the SCP-1 treated cells using Trifast (Peqlab, Erlangen, Germany) according to the manufacturer´s protocol and quantified using a spectrophotometer (Omega plate reader, BMG Labtech GmbH, Germany). cDNA was synthesized using the First Strand cDNA Synthesis Kit from 2500 ng total RNA (Fermentas St, Leon-Rot, Germany). Afterwards, semi-quantitative RT-PCR was performed from the 10 ng cDNA template using KAPA2G Fast Ready Mix (Peqlab, Erlangen, Germany). Primers and PCR conditions were previously optimized with increasing amounts of cDNA, in order to analyzed the PCR product obtained from the logarithmic phase. Primer sequences and PCR conditions are shown in Table 2. GAPDH was used as an internal control for normalization. PCR products were resolved using a 1.5% agarose gel, with ethidium bromide for visualization. Densitometric analysis was performed using the ImageJ software.

### 4.9. Statistic Analysis

Results were represented as bar diagrams (mean ± SEM) of at least three independent experiments (biological replicates, *N* ≥ 3) measured as triplicate or more (technical replicates, *n* ≥ 3). The comparison of multiple groups was done using the Kruskal–Wallis H-test, followed by the Dunn’s multiple comparison test. The Mann–Whitney U-test (two-sided) was used to compare two single groups with each other. Statistical analysis was performed using the GraphPad Prism Software (Version 5, El Camino Real, CA, USA). *p* < 0.05 was considered as the minimum level of statistical significance.

## 5. Conclusions

In summary, our study demonstrated for the first time that TGF-β signaling was impaired in CSE-exposed MSCs. Moreover, the disruption of primary cilia with CSE and CH affected proper TGF-β signal transmission thought the cell. Therefore, TGF-β signaling dysregulation contributes in part to the adverse effects observed in MSC migration, proliferation, and differentiation, which could explain the effect of endochondral ossification, and consequently, impair or delay long bone fracture healing in smokers.

## Figures and Tables

**Figure 1 ijms-20-02915-f001:**
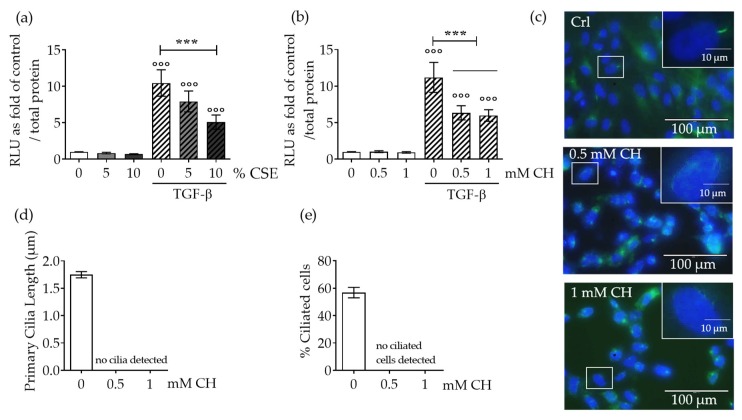
CSE exposure decreased TGF-β signaling by disrupting MSC primary cilia. Single-cell-derived human mesenchymal stem cell line (SCP-1 cells (*N* = 4, *n*  =  8)) infected with an adenoviral Ad5-CAGA9-MLP-Luciferase reporter constructs (Smad2/3 reporter) were exposed overnight, either with or without Cigarette smoke extract (CSE) (**a**; 5–10%) or chloral hydrate (CH) (**b**; 0.5–1 mM). Next, cultures were incubated with recombinant human transforming growth factor beta one (rhTGF-β1) 10 ng/mL for 48 h, and luciferase activity was measured in cell lysates. The results were normalized to total protein content and expressed as relative luminesce units (RLU). Results represent mean ± standard error of the mean (SEM). Statistical significance was determined by the Kruskal–Wallis H test, followed by Dunn’s post-test. Significance was established as *** *p* < 0.001 compared to TGF-β-treated cells and °°° *p* < 0.001 compared to untreated cells. (**c**) Representative immunostaining images of SCP-1 cells stained for acetylated α-tubulin (green), and nuclei (blue), after CH exposure. (**d**) Primary cilia length quantification of SCP-1 cells treated with and without CH. (**e**) Percentage of ciliated SCP-1 cells following CH treatment.

**Figure 2 ijms-20-02915-f002:**
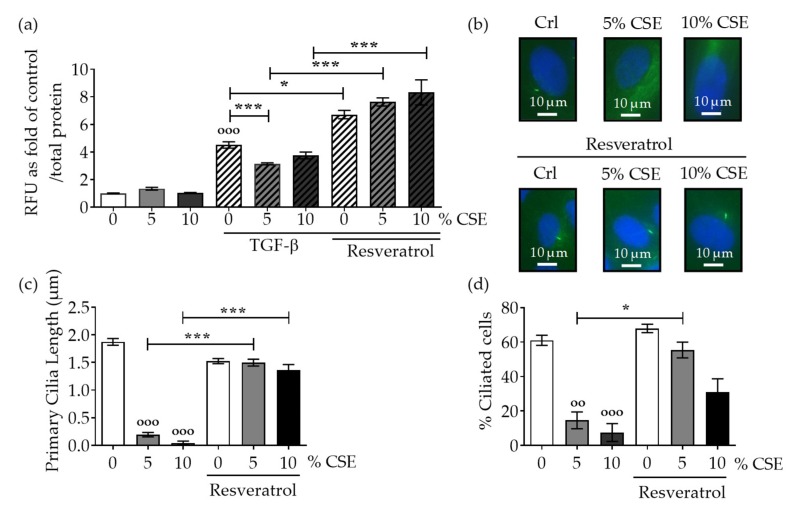
Resveratrol preserves the primary cilia structure from CSE and reestablishes TGF-β signaling. SCP-1 cells (*N*  =  3, *n*  =  6) infected with Ad5-CAGA9-MLP-Luciferase reporter constructs (Smad2/3 reporter) were co-incubated overnight, either with or without CSE (5–10%) and resveratrol (1 µM). Next, cultures were incubated with rhTGF-β1 (10 ng/mL) for 48 h, and (**a**) luciferase activity was measured in cell lysates. The results were normalized to total protein content and expressed as relative luminesce units (RLU). Results represent mean ± standard error of the mean (SEM). Statistical significance was determined by the Kruskal–Wallis H test, followed by Dunn’s post-test. Significance was established as * *p* < 0.05, *** *p* < 0.001 compared to TGF-β-treated cells and °°° *p* < 0.001 compared to untreated cells. (**b**) Representative immunostaining images of SCP-1 cells stained for acetylated α-tubulin (green), and nuclei (blue), after incubation with CSE and resveratrol. (**c**) Primary cilia length quantification of SCP-1 cells treated with and without CSE and resveratrol. (**d**) Percentage of ciliated SCP-1 cells following resveratrol treatment.

**Figure 3 ijms-20-02915-f003:**
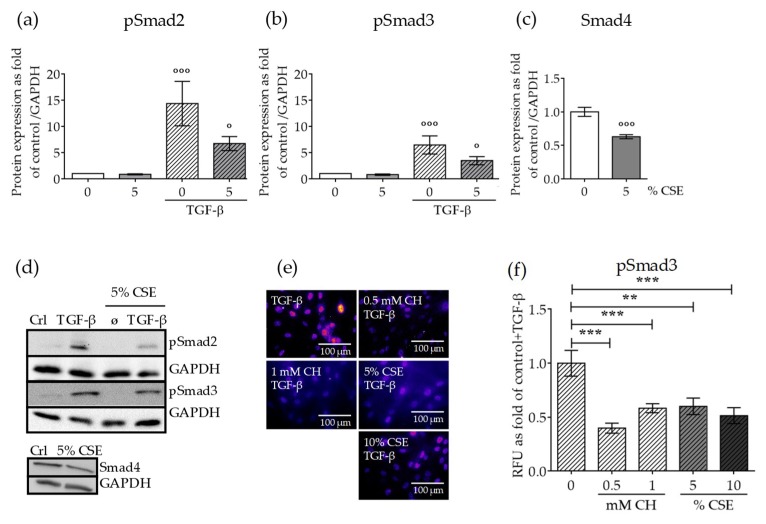
CSE exposure affected protein expression levels of canonical TGF-β signaling mediators and their nuclear translocation. SCP-1 cells (*N*  =  3, *n*  =  3) were exposed to CSE (5%) twice a week. After 14 days, the cells were treated with rhTGF-β1 (10 ng/mL) for 1 h. Protein expression of phospho-Smad2 (**a**), phospho-Smad3 (**b**), and Smad4 (**c**) was measured in cell lysates by Western blot and normalized to Glyceraldehyde 3-phosphate dehydrogenase (GAPDH). (**d**) A representative Western blot of the measured proteins is shown. SCP-1 cells (*N*  =  3, *n* = 3) were treated overnight with or without CSE (5–10%) or CH (0.5–1 mM). Next, cultures were incubated with rhTGF-β1 (10 ng/mL) for 24 h and then stained for Smad3. (**e**) Representative immunostaining images of nuclear localization of Smad3. The immunofluorescence signal was pseudocolored for better visualization using the fire tool in ImageJ. (**f**) Quantification of Smad3 nuclear translocation was performed with ImageJ. The results are expressed as mean ± SEM. Statistical significance was determined by the Kruskal–Wallis H, test followed by Dunn’s post-test. Significance was established as ** *p* < 0.01 or *** *p* < 0.001 compared to TGF-β and ° *p* < 0.05 or °°° *p* < 0.001 compared to untreated cells.

**Figure 4 ijms-20-02915-f004:**
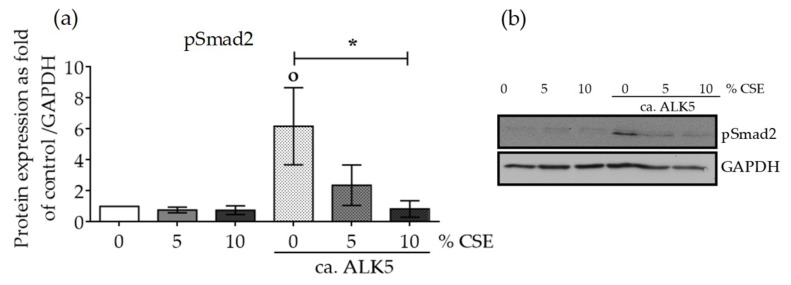
CSE reduced the protein expression of phospho-Smad2, despite a constitutively active TGF-β receptor type I (ca. ALK5). SCP-1 cells (*N*  =  3, *n*  =  3) infected with Ad5-ca. ALK5 were treated with CSE (5–10%). After 24 h, the total protein expression of phospho-Smad2 (**a**) was measured in cell lysates by Western blot and normalized to GAPDH. (**b**) A representative Western blot is shown. The results are expressed as mean ± SEM. Statistical significance was determined by the Kruskal–Wallis H test, followed by Dunn’s post-test. Significance was established as * *p* < 0.05 compared to circa ALK5-infected cells and ° *p* < 0.05 compared to untreated cells.

**Figure 5 ijms-20-02915-f005:**
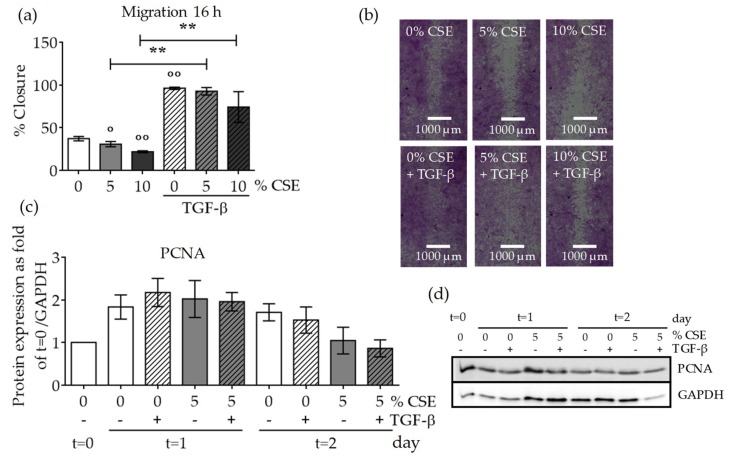
CSE exposure decreased MSC migration and proliferation. In order to investigate the effect of CSE on SCP-1 cell migration, a scratch assay was performed. SCP-1 cells (*N* ≥ 3, *n* ≥ 3) were co-incubated with CSE (5–10%) and rhTGF-β1 (10 ng/mL). Wound closure was determined from microscopic pictures (equation: (100 − wound area at 16 h/wound area at 0 h) × 100) with ImageJ software. (**a**) SCP-1 cell migration after 16 h. (**b**) Representative migration pictures. Cells were visualized with sulforhodamine B (SRB) staining. (**c**) Normalized proliferating cell nuclear antigen (PCNA) protein expression in SCP-1 cells at day 0 (*t* = 0) and after 24 h (*t* = 1) or 48 h (*t* = 2) stimulation with rhTGF-β1 (10 ng/mL) and with or without CSE (5% *v/v*). (**d**) A representative PCNA Western blot. The results are expressed as mean ± SEM. Statistical significance was determined by the Kruskal–Wallis H test, followed by Dunn’s post-test. Significance was established as ** *p* < 0.01 compared to TGF-β1-treated cells and ° *p* < 0.05 or °° *p* < 0.01 compared to untreated cells.

**Figure 6 ijms-20-02915-f006:**
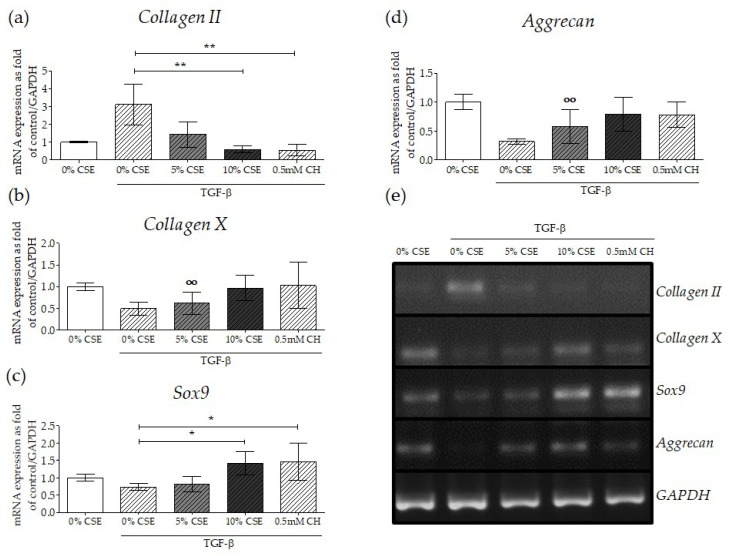
Disruption of TGF-β signaling with CSE-mediated primary cilia disruption affected MSC chondrogenic differentiation. In order to evaluate the gene expression of chondrocyte markers under impaired TGF-β signaling, SCP-1 cells (*N* = 3, *n* = 2) were differentiated with CSE (5–10%) and rhTGF-β1 (10 ng/mL) for 14 days. Gene expression analysis was performed with semi-quantitative RT-PCR from 10 ng cDNA. The graph represents gene expression, normalized to the *GAPDH* (housekeeping gene), of (**a**) *Collagen II*, (**b**) *Collagen X*, (**c**) *Sox9*, and (**d**) *Aggrecan*. The results are expressed as mean ± SEM. Statistical significance was determined by the Kruskal–Wallis H test, followed by Dunn’s post-test. Significance was established as ** *p* < 0.01 or * *p* < 0.05 compared to TGF-β1-treated cells or °° *p* < 0.01 compared to untreated cells. (**e**) A representative semi-quantitative reverse-transcription polymerase chain reaction (RT-PCR) gel picture.

**Table 1 ijms-20-02915-t001:** Antibodies used in Western blot and immunofluorescence staining.

Antibody	Catalog No.	Company	Dilution
phospho-Smad2	3108	Cell Signaling	1:1000
phospho-Smad3	9520	Cell Signaling	1:1000
Smad4	9515	Cell Signaling	1:1000
PCNA	ab92522	Abcam	1:1000
HRP antirabbit IgG	sc-2004	Santa Cruz	1:10000
Smad3	9523	Cell Signaling	1:50
Alexa 488 antirabbit IgG	A21206	Invitrogen	1:1000
Acetylated αTubulin (6-11b-1)	sc-23950	Santa Cruz	1:100
Alexa 488 antimouse IgG	A10667	Invitrogen	1:1000

**Table 2 ijms-20-02915-t002:** Primer sequences and PCR conditions for the investigated genes.

Gene	Accession Number	Forward Primer (5′–3′)	Reverse Primer (3’–5’)	Product Length (bp)	Annealing Temperature (°C)	Cycles. (N°)
*Aggrecan*	NM_001135.3	CTTGGACTTGGGCAAACTGC	CACTAAAGTCAGGCAGGCCA	143	60	35
*Collagen II*	NM_001844.4	TGGATGCCACACTCAAGTCC	GCTGCTCCACCAGTTCTTCT	254	60	35
*Collagen X*	NM_000493.3	AAACCTGGACAACAGGGACC	CGACCAGGAGCACCATATCC	124	60	35
*SOX9*	NM_000346.3	GAAGGACCACCCGGATTACA	GCCTTGAAGATGGCGTTGG	120	60	35
*GAPDH*	NM_002046.4	GTCAGTGGTGGACCTGACCT	AGGGGTCTACATGGCAACTG	420	56	30

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
