# Peer review of "Cigarette Smoke Induces the Risk of Metabolic Bone Diseases: Transforming Growth Factor Beta Signaling Impairment via Dysfunctional Primary Cilia Affects Migration, Proliferation, and Differentiation of Human Mesenchymal Stem Cells"

_ijms, 2019, doi:10.3390/ijms20122915_

Round 1

Reviewer 1 Report

The authors analyzed the effects of cigarette smoke extract (CSE) on TGF-beta signaling and primary cilia structure in an immortalized human mesenchymal stem cell line (SCP-1). CSE exposure decreased TGF-beta signaling, which might explain delayed fracture healing in smokers. Unfortunately, the authors’ interpretation of data is not scientifically convincing. There are numerous problems in the manuscript including those outlined blow.

Major problems:

1.     Specific conclusions such as “CSE exposure reduced the activation of TGF-beta modulators by constitutive activation of TGF-beta receptor type I (ALK5)” (Abstract) is not at all supported by experimental data.

2.     The authors performed experiments by using CSE, which is a mixture of various components including nicotine and cotinine and can have diverse effects. Therefore, “CSE exposure decreased TGF-beta signaling by disrupting MSC primary cilia” (Figure 1) is not fully supported by data. This is because effects of CSE or CH on TGF-beta signaling and primary cilia structure can be independent.  Similarly, “These results suggested that defective primary cilia lead to aberrant cell signaling coordination under smoking conditions” (Line 137, Figure 2) is not directly supported by data. Without rescue experiments, it is difficult to draw their conclusion “via dysfunctional primary cilia”.

Minor points:

1.     Fracture healing is a complex process involving not only mesenchymal cells but osteoclasts. The authors should properly describe limitations of this study not considering osteoclasts.

2.     The authors only used human immortalized mesenchymal stem cell line (SCP-1), which is not enough. The authors should test bone marrow derived MSCs in addition to SCP-1 cells.

3.     Figure 1(c), Percentage of ciliated cells and cilia length should be quantified.

4.     Figure 2(e) legend lacks explanation. Figure 2(f) should be quantified.

5.     Figure 4(a), Is the CSE effects on %Closure significant? The authors should perform statistical test. Figure 4(c) also needs statistical tests.

6.     Figure 4 needs time course study to demonstrate “delayed MSC chondrogenic differentiation”.

7.     4.4. Ad5-CAGA9-MLP-Luciferase and Ad5-caALK5 need references.

8.     Reference 24 and 65 are identical (duplicated).

9.     The following reference should be discussed.

Deren ME, Yang X, Guan Y, Chen Q. Biological and Chemical Removal of Primary Cilia Affects Mechanical Activation of Chondrogenesis Markers in Chondroprogenitors and Hypertrophic Chondrocytes. Int J Mol Sci. 2016;17(2):188. doi: 10.3390/ijms17020188.

10.  The link to the authors previous publications using CSE and SCP-1 cells is not well explained and unclear. Please discuss them better.

Author Response

Please find the detailed reviewered response in the attached PDF

Reviewer 2 Report

The authors claim that ‘’ smokers frequently have decreased of TGFB “, this result seems to be observed in a specific case, after a long bone fracture. Usually an upregulation of TGF-β legends are observed in major pulmonary diseases, including pulmonary fibrosis, emphysema and lung. Pathologically activated TGF-β signaling in relation to smoking are involved in emphysema and fibrosis, TGF-β is a key player in fibrotic processes, acting on macrophages, fibroblasts and alveolar epithelial cells. TGF-β regulates multiple cellular processes in this disease such as growth suppression of epithelial, alveolar epithelial cell differentiation, fibroblast activation, and extracellular matrix organization.
Therefore, it is difficult to reviewer believes that CSE can decrease TFG signaling in MSCs cell. Then, different points need to be clarified to persuade the reviewer.
CSE can be toxic in many kinds of cell as such epithelial cells, macrophages. The review would like to know if the authors did toxify test of CSE in MCS cells. The authors did not show if CSE 5% or 10% has toxic effects on stem cell. Now is difficult to know if impairment of TGF-B signaling is due to the toxic effect of CSE.

In line, 119 the author claim that “SPC1 cells were exposed to CSE 5%, and after 14 days and TGF-b signaling was induced”. The extract of cigarette smoke is a preparation with timeless activities, how do the authors guarantee these activities for 14 days? The reviewer would like to know why only after 14 days of CSE exposure the cells were exposed to TGF-β, and not 24h as in figure 1? The reviewer are not sure that after 14 days it possible observed the effect of CSE?  In this case,  the CSE was prepared every 2 days? Please clarify these points.

In addition, protein phosphorylation signaling can be observed in short periods of exposure. Why was 14-days exposure required to decrease the expression of TGF-b signaling proteins in Figure 2? However, we can observe a change in primary cilia in only 48h (Figure 1). Please clarify this point.

Author Response

(The authors gave the same response as above.)

Reviewer 3 Report

The authors describe a well performed study using different experimental set-ups to investigate the effect of smoking on TGF-ß1 signaling in human MSCs. The study provides important new information and explanation for the bone healing impairment seen in smokers. The discussion is very detailed and references important studies.

I have only some comments:

Due to the fact that the M&M part is at the end of the manuscript, details on the ells and exp. Set-up must briefly be mentioned in the results.

Line 92: please give information on the SPC-1 cells.

Line 119: why was 5% CSE used and not 10%, which showed a significant effect in the previous experiment (smad2/3 reporter activity)?

Figure 4c: no significant changes are indicated. In line 187, however, a sig. decrease is described.

Line 221ff, 338ff: the conclusion overestimates the in vitro results and should be formulated more carefully.

Fig. 5 Does this culture condition really induces chondrogenic differentiation? Only expression of Col II was increased due to TGF stimulation, but no other chondrogenic factor. Please provide data that culturing in chondrogenic medium results in chondrogenic differentiation of the SCP-1 cells.

Please always write TGF-β1 instead of only TGF-β.

Author Response

(The authors gave the same response as above.)

Round 2

Reviewer 2 Report

Authors respond satisfactorily to the reviewer's question